biomechanics

XROMM, trout, frogfish, intervertebral joint, craniovertebral, cranial elevation

**Author for correspondence:**
Ariel L. Camp
e-mail: ariel.camp@liverpool.ac.uk

# A neck-like vertebral motion in fish

Ariel L. Camp[1,2]

[1]Department of Musculoskeletal and Ageing Science, Institute of Life Course and Medical Sciences, University of Liverpool, Liverpool, UK
[2]Department of Ecology, Evolution and Organismal Biology, Brown University, Providence, USA

 ALC, 0000-0002-3355-4312

Tetrapods use their neck to move the head three-dimensionally, relative to the body and limbs. Fish lack this anatomical neck, yet during feeding many species elevate (dorsally rotate) the head relative to the body. Cranial elevation is hypothesized to result from the craniovertebral and cranial-most intervertebral joints acting as a neck, by dorsally rotating (extending). However, this has never been tested due to the difficulty of visualizing and measuring vertebral motion *in vivo*. I used X-ray reconstruction of moving morphology to measure three-dimensional vertebral kinematics in rainbow trout (*Oncorhynchus mykiss*) and Commerson's frogfish (*Antennarius commerson*) during feeding. Despite dramatically different morphologies, in both species dorsoventral rotations extended far beyond the craniovertebral and cranial intervertebral joints. Trout combine small (most less than 3°) dorsal rotations over up to a third of their intervertebral joints to elevate the neurocranium. Frogfish use extremely large (often 20–30°) rotations of the craniovertebral and first intervertebral joint, but smaller rotations occurred across two-thirds of the vertebral column during cranial elevation. Unlike tetrapods, fish rotate large regions of the vertebral column to rotate the head. This suggests both cranial and more caudal vertebrae should be considered to understand how non-tetrapods control motion at the head–body interface.

## 1. Introduction

All vertebrates need to control the motion between the head and body. Tetrapods do this with the neck: a region of specialized vertebrae between the cranium and pectoral girdle, which functions to move the head three-dimensionally relative to the body and limbs [1,2]. In ray-finned fish ('fish' hereafter), the anatomy is quite different. In most fish, the pectoral girdle articulates directly with the cranium (figure 1) and these joints determine the cranial–pectoral mobility [3]. A separate articulation links the vertebral column and the skull [4]. While the cranial-most vertebrae in some fish form morphologically [5] and developmentally [6] distinct cervical regions, no study has yet determined if these regions function like a tetrapod neck.

One important neck-like function is rotating the head dorsoventrally in the sagittal plane, relative to the body [4,7]. The vertebral column of fish is well known to flex laterally during swimming [8–10] but only small amounts of dorsoventral and axial flexion are hypothesized in the caudal intervertebral joints [6]. Dorsoventral flexion has never been directly measured across the cranial vertebrae. Yet during feeding, many fish elevate (dorsally rotate) the neurocranium relative to the body. Cranial elevation is widely used to expand the mouth cavity [11] and can only be produced by active shortening of the epaxial muscles to dorsally rotate the craniovertebral and/or intervertebral joints. Epaxial muscle activation and shortening is well documented in fish [12,13] and can occur over large regions extending halfway down the body [13,14]. Which intervertebral joints the epaxial muscles rotate and by how much remains unknown.

The most common hypothesis is that fish rely on rotations of the craniovertebral and first 2–5 postcranial intervertebral joints [4,15,16] to elevate the head—analogous to the tetrapod neck. In support of this hypothesis, these

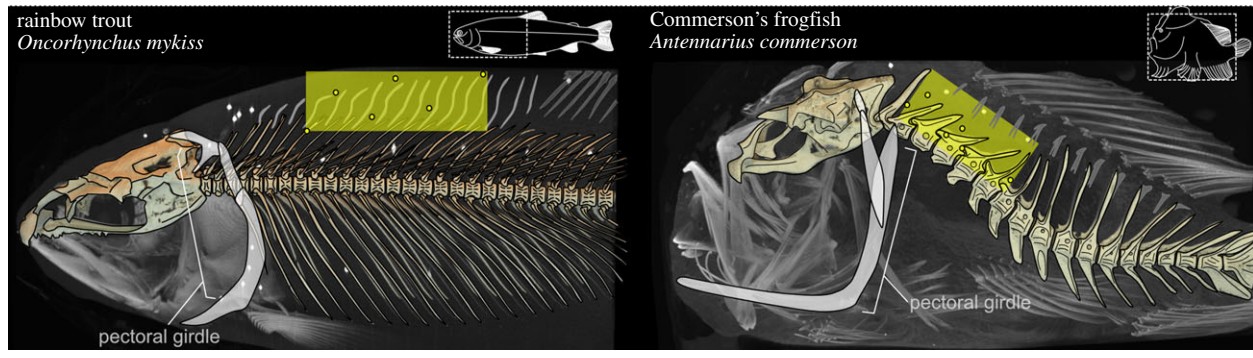

**Figure 1.** Craniovertebral skeleton of trout and frogfish. Renderings of a maximum intensity projection and bone models show the intramuscular markers (yellow, outlined circles) of the body plane (yellow rectangle) and additional markers (white circles) not analysed in this study. Insets show the body shape of each species, and the region of the vertebral column studied (dashed box). (Online version in colour.)

cranial-most vertebrae are often anatomically distinct [4,5] and near the centre of cranial rotation [15]. Alternatively, cranial elevation could be produced by rotation of many intervertebral joints across large regions of the vertebral column, as during lateral body flexion for swimming, for examples see [8,9]. These hypotheses have not been tested, as intervertebral joint rotations particularly in the sagittal plane are difficult to measure externally.

I used X-ray reconstruction of moving morphology (XROMM) [17,18] to measure three-dimensional motions of the neurocranium and vertebral column during feeding in fish representing two extremes of vertebral morphology and cranial kinematics. Rainbow trout (*Oncorhynchus mykiss*) have 62 vertebrae and moderate cranial elevation [19]. While the first two vertebrae form a cervical region [5], all the vertebrae are relatively small, with a similar shape and little bony overlap at the intervertebral joints (figure 1; electronic supplementary material, figure S1). By contrast, Commerson's frogfish (*Antennarius commerson*) have 18 large vertebrae that vary dramatically in their shape and size, and extremely large cranial elevation [20]. The first two vertebrae are morphologically distinct and not unlike an atlas–axis complex (figure 1; electronic supplementary material, figure S1).

## 2. Material and methods

### (a) Animal models

Three female, adult, rainbow trout (*Oncorhynchus mykiss*) were obtained from Kilnsey Trout Farm (Skipton, UK) with mass and standard lengths of 776 g and 345 mm (OM1), 770 g and 340 mm (OM2), and 998 g and 375 mm (OM3). Three adult Commerson's frogfish (*Antennarius commerson*) were obtained from Tropic Marine Centre, with masses and standard lengths of 368 g and 192 mm (AC1), 429 g and 205 mm (AC2), and 577 g and 230 mm (AC4). All fish were housed at the University of Liverpool in individual tanks.

### (b) X-ray video and morphological data collection

Fish were anaesthetized with buffered tricaine methanesulfonate (MS-222), and 0.5–0.8 mm tantalum beads (X-medics) were coated in an analgesic (lidocaine) and implanted in the neurocranium and epaxial muscles [13]. Up to six epaxial markers were implanted midsagittally (trout; three deep, three superficial) or laterally (frogfish; 1–3 left-lateral, 1–3 right-lateral) to form a body plane (figure 1), a fish-based reference for measuring cranial elevation [13]. All individuals recovered fully and resumed normal feeding behaviours within 24 h.

Biplanar, high-speed X-ray video was recorded of each fish feeding on pellets and mealworms (trout) or dead anchovies and shrimp (frogfish) in a 10 × 36 × 41 cm corridor of their tank. X-ray machines (Imaging Systems and Service) generated images, which were recorded by two Phantom cameras (M120, Vision Research) at 1024 × 1024 resolution, 500 fps and 1/1000 shutter speed (trout) or 1000 fps and 1/4000 shutter speed (frogfish). At least four strikes were recorded from each fish, but only strikes with at least 5° of cranial elevation were analysed. This included all 18 recorded frogfish strikes and 20 of the 30 trout strikes. All frogfish strikes were successful, and only two of the analysed trout strikes were unsuccessful.

Fish were induced and euthanized with an overdose of buffered MS-222 and frozen before computed tomography (CT) scans were taken (512 × 512 pixel resolution, 0.172 mm slice thickness) on a Quantum GX microCT scanner (PerkinElmer) at the University of Liverpool Centre for Preclinical Imaging. Polygonal mesh models of the tantalum markers, neurocranium, and each vertebra were created in Horos (v 3.3.6, horosproject.org) and Dragonfly (v 2020.2, Object Research Systems Inc.).

### (c) X-ray reconstruction of moving morphology animation

Three-dimensional motion of the bones and body plane were reconstructed with a combination of marker-less [18] and marker-based XROMM [17]. X-ray videos were undistorted, the three-dimensional space calibrated, and neurocranium and body plane markers tracked using XMALab (v. 1.53) [21]. Rigid body transformations were calculated for the neurocranium and body plane, filtered with a low-pass Butterworth (60 Hz cut-off for trout; 100–200 Hz for frogfish) and used to animate these models in Maya (v2020, Autodesk). Marker tracking precision was less than 0.1 mm across all trout strikes, calculated as the mean standard deviation of the distance between neurocranium markers [17]. Precision could not be calculated from frogfish neurocranium markers, as X-ray and CT images showed these were in the soft tissues surrounding the neurocranium. All frogfish vertebrae and the cranial 24–27 trout vertebrae were animated in Maya with Scientific Rotoscoping [18]. For frogfish, marker-based neurocranium animations were augmented or replaced by rotoscoping.

### (d) Skeletal kinematics

Cranial and vertebral kinematics were measured using the oRel and Axes tools from the XROMM_MayaTools package (https://bitbucket.org/xromm/xromm_mayatools). Joint coordinate systems (JCSs) were placed at the craniovertebral and intervertebral joints. Each JCS consisted of two anatomical coordinate systems (ACSs) with the *z*-axis oriented left-to-right laterally (describing

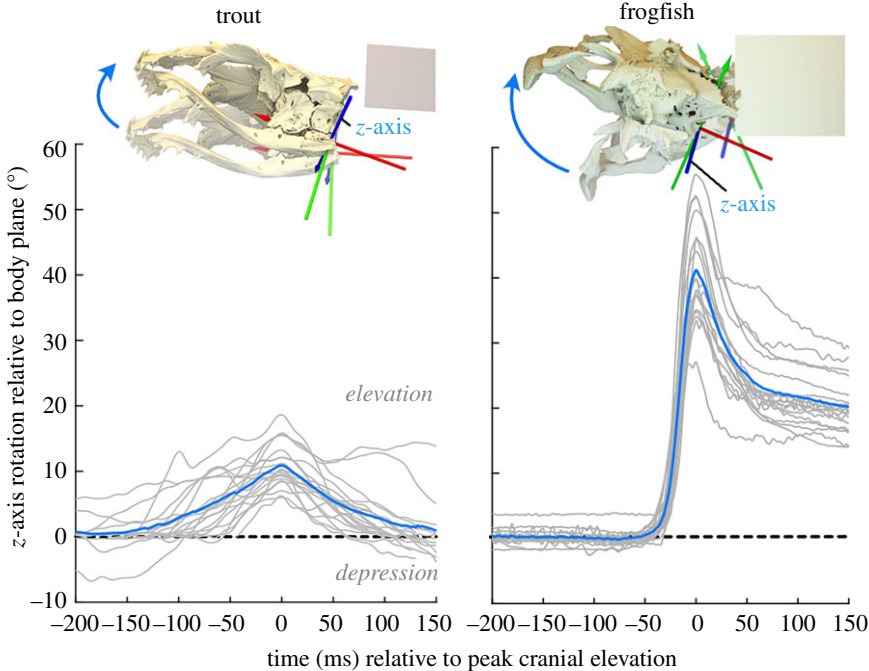

**Figure 2.** Cranial elevation during feeding strikes. Top: the three-dimensional joint coordinate system for measuring cranial elevation, with the initial (semi-transparent) and maximally elevated (opaque) cranial position relative to the body plane (beige rectangle). Curved arrows indicate positive z-axis rotation. Bottom: cranial elevation over time for each strike (light grey lines) and the mean cranial elevation at every time step (dark blue line). (Online version in colour.)

dorsoventral rotation in the sagittal plane and lateral translation), the $y$-axis oriented dorsoventrally (lateral rotation and dorsoventral translation) and the $x$-axis craniocaudally (axial rotation and craniocaudal translation). One ACS was attached to the cranial vertebra and the other to the caudal vertebra of each intervertebral joint. Joint motions were measured following a $zyx$ rotation order. Dorsoventral intervertebral joint rotations were measured as $z$-axis rotations of each vertebra relative to its cranial neighbour. These $z$-axis rotations were multiplied by −1 so dorsal rotation corresponded to positive $z$-axis rotation. Cranial elevation was measured with a JCS at the craniovertebral joint, consisting of ACSs (same orientation as above) attached to the neurocranium and the body plane (figure 2). Positive $z$-axis rotations of this JCS corresponded to cranial elevation.

To visualize the curvature of the vertebral column in the sagittal plane, the dorsoventral translations of the neurocranium and each vertebra were recalculated relative to an ACS attached to the caudal-most vertebra (same orientation as above). For trout, this was vertebra 24 (OM1), 27 (OM2) or 25 (OM3) and for frogfish this was vertebra 18. The translations of virtual landmarks at the craniovertebral joint and each vertebral centra were calculated relative to the caudal ACS (figure 3$a$).

## 3. Results

Both species consistently elevated (dorsally rotated) the neurocranium relative to the body plane during feeding strikes (figure 2). Trout reached peak cranial elevations of up to 18.6° (mean of 10.9 ± 0.76° s.e.) over 50–150 ms (figure 2). Frogfish elevated the neurocranium by up to 56° (mean of 41 ± 2° s.e.) over 30–50 ms (figure 2). Capturing non-elusive food items is unlikely to have elicited maximum performance, so these are conservative measures of the cranial and vertebral motions these fish are capable of.

Dorsoventral vertebral motion extended far beyond the craniovertebral joint in trout and frogfish (figure 3). Trout started the strike with the vertebral column nearly horizontal,

then extended the anterior vertebrae dorsally at peak cranial elevation (figure 3$a$). The magnitude varied considerably, but in nearly all strikes the neurocranium moved dorsally relative to the caudal vertebra (figure 3$b$). The six cranial-most vertebrae had the greatest magnitude of translation per centrum, with magnitudes decreasing caudally (figure 3$c$).

Frogfish began strikes with a C- or S-shaped vertebral column: the cranial vertebrae flexed and the more caudal vertebrae extended (figure 3$a$). During the strike, the more cranial vertebrae extended dorsally to become straight or curve dorsally. The more caudal vertebrae either straightened—flexing ventrally—or extended dorsally (figure 3$b$). The cranial two vertebrae of frogfish had the greatest dorsal translation per centrum, while the 3–7th most cranial vertebrae had the greatest ventral translation per centrum. The more caudal vertebrae (8–12th postcranial) either extended as the head moved dorsally or flexed with ventrally head motion (figure 3$b$; electronic supplementary material, figure S2).

Trout produced cranial elevation through small rotations of many intervertebral joints. The dorsoventral rotation of each intervertebral joint was always less than 7° (usually less than 3°) and highly variable across strikes and individuals (figure 4$a$; electronic supplementary material, figure S3). The largest rotations often occurred over the craniovertebral and first two intervertebral joints, but no single joint was sufficient to produce cranial elevation.

Frogfish underwent large dorsal and ventral joint rotations to reshape the vertebral column as the head moved. The craniovertebral and first intervertebral joint had the greatest dorsal rotations (mean ± s.e. of 18 ± 1° and 24 ± 2°, respectively), and these two joints could have generated greater than 80% of maximum cranial elevation. Rotations at the remaining joints were usually less than 10°, including substantial ventral rotation over the 7–9th intervertebral joints, and smaller dorsal and ventral rotations of the caudal (10–12th) joints (figure 4$b$).

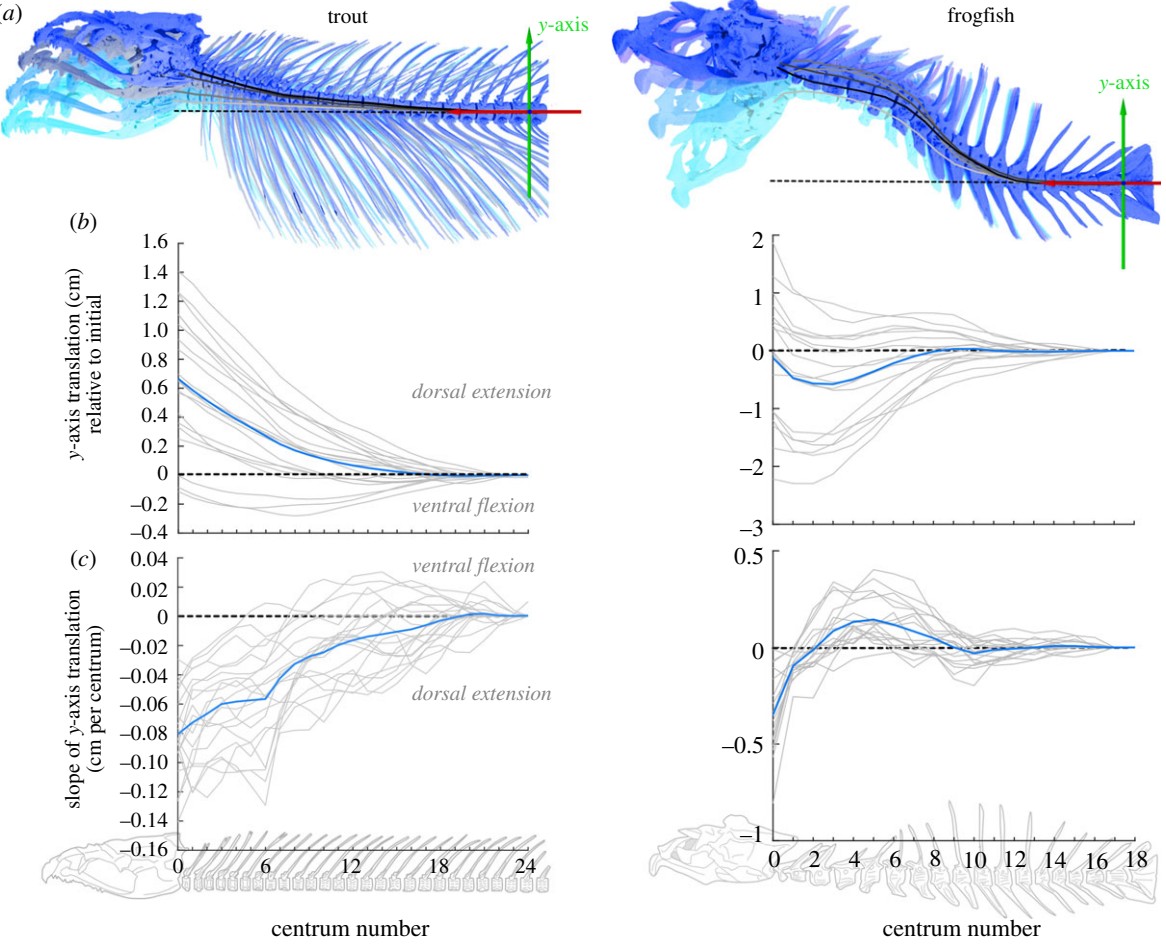

**Figure 3.** Dorsoventral curvature of the vertebral column in trout and frogfish. (*a*) Positions of the neurocranium and vertebrae at four frames of a sample strike, from initial (lightest blue) to maximum cranial elevation (darkest blue). Lines show the dorsoventral (*y*-axis) centrum translations relative to the caudal ACS, from the initial position (lightest grey) to maximum cranial elevation (black). The dashed line shows zero translation. (*b*) Dorsoventral (*y*-axis) centrum translations, relative to their initial value for each strike (light grey lines) and the mean at each centrum across all strikes (dark blue lines). (*c*) *Y*-axis translation per centra for each strike (light grey lines) and the mean for each centrum (dark blue lines). Centrum 0 indicates the neurocranium. (Online version in colour.)

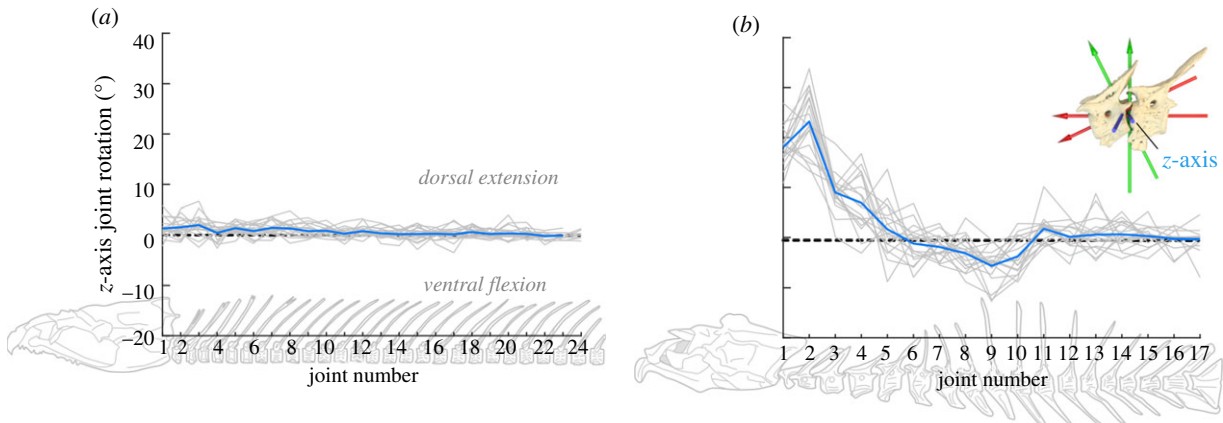

**Figure 4.** Dorsoventral rotation in the sagittal plane of intervertebral joints at peak cranial elevation. (*a*) Trout and (*b*) frogfish rotations relative to their initial value for each strike (light grey lines) and the mean rotation for each joint (dark blue lines). Joint 1 is the craniovertebral joint. (Online version in colour.)

## 4. Discussion

Trout and frogfish used large regions of the vertebral column—extending far beyond the craniovertebral joint and pectoral girdle—as a functional neck to rotate the head dorsoventrally. By extending up to a third of their vertebral column, trout produced 10–15° of cranial elevation despite small (usually less than 3°) intervertebral joint rotations.

Frogfish concentrated the largest rotations over the cranial-most joints, but the entire cranial two-thirds of the vertebral column rotated dorsoventrally during cranial elevation. Rather than relying on the cranial-most vertebrae to rotate the head, like tetrapods, fish use large regions of the vertebral column for both feeding and swimming.

Trout and frogfish demonstrate two ways fish can elevate the head relative to the body. Trout combined relatively

small dorsal rotations across the anterior 10–15 intervertebral joints to elevate the neurocranium (figure 4), while the whole fish moved in the water column to position its head near the food. In frogfish, large rotations of the cranial-most 2–3 joints were sufficient to elevate the neurocranium, while dorsoventral rotations of the remaining vertebrae corresponded with the position the head (figure 3; electronic supplementary material, figure S2). Frogfish did not move the body during the strike, but extended and rotated the vertebral column towards the food. These differences may reflect the vertebral morphology of each species. Trout vertebrae are relatively small, similarly shaped, with little bony articulation, while frogfish vertebrae are relatively large, anatomically varied, and many have well-developed bony articulations (figure 1; electronic supplementary material, figure S1). However dorsoventral rotations were not restricted to the two cervical [5] vertebrae in trout or to a single vertebral morphology in frogfish (figures 3 and 4). Thus, the vertebral column of fish seems unlikely to have a simple relationship between anatomical and functional regionalization.

The epaxials are the only muscles that can dorsally rotate the neurocranium and vertebral column [16] and I expect they actively shorten during cranial elevation [14,22]. But epaxial activity may be extending the vertebrae to elevate the neurocranium, or elevating the neurocranium to extend the vertebrae as in seahorses [23], or both. It is also unclear how vertebral motion is controlled as the epaxial muscle segments (myomeres) have complex shapes and span multiple intervertebral joints, although activation and shortening can vary within a single myomere [24]. This may be particularly challenging in frogfish where the magnitude and direction of vertebral rotation vary considerably (figure 4). Epaxial morphology has not been described in frogfish, but in other trout species myomere shape varies craniocaudally [25]. Further research is needed to examine how this corresponds to dorsoventral vertebral rotation.

The external fluid forces of suction feeding that 'suck' the fish towards the food could also cranially translate and straighten the vertebral column (figure 3). This is most likely to be relevant in frogfish, where extremely large and rapid mouth expansion will produce strong suction flows [20], while fish held the body against the tank walls. Suction flows are likely powered primarily by the epaxial muscles, suggesting they contribute to both the internal and external forces acting on the head and vertebrae.

I expect dorsoventral rotation over many intervertebral joints is common among ray-finned fish. Some dorsoventral rotation is essential for cranial elevation, which is used by many species [26] to expand the mouth during feeding or move the head relative to the body to control mouth position [27]. Trout and frogfish show large regions of dorsoventral rotation are not limited to a single morphology, feeding behaviour or phylogenetic group. And since fish actively shorten large regions of the axial muscles during feeding [13], using a similar portion of the axial skeleton would not be surprising. Alternatively, some intervertebral joints may also contribute to feeding through reduced rotation. While frogfish rely on extremely large rotation of the cranial-most intervertebral joints (figure 4), in other species with large and fast cranial elevation the first 1–4 vertebrae are actually fused to the neurocranium, e.g. *Luciocephalus pulcher* [28] and *Aulostomus maculatus* [29]. One function of these fused vertebrae may be to resist the high forces resulting from epaxial muscles acting on the neurocranium during powerful cranial elevation, while dorsoventral rotation occurs over more caudal joints . Therefore, dorsoventral rotation and feeding—not just lateral flexion and swimming—need to be considered in studying the biomechanics and evolution of fish vertebrae. But dual-function vertebrae are probably not universal, for example, the cranial elevation of seahorses is achieved primarily by craniovertebral joint rotation [23].

This study raises new questions about how non-tetrapod vertebrates may use neck-like, dorsoventral rotations to move the head. Cranial elevation is hypothesized to be an ancestral mouth-opening mechanism for gnathostomes [30], so dorsoventral mobility at the craniovertebral interface may also be an ancient trait of jawed vertebrates. In living sharks and rays cranial elevation is often minimal [31], although the pectoral girdle of sharks is separated from the neurocranium [32]. Living lobe-finned fishes can elevate the neurocranium [33] and one extinct sarcopterygian fish has a functional neck, based on its craniovertebral joint morphology [34]. This suggests one neck-like function—dorsoventral rotation between the head and body—could have been present in fully aquatic fishes before the origin of tetrapods. To test this, research is needed on the dorsoventral mobility of both the craniovertebral and intervertebral joints in extant and fossil fishes.

## 5. Conclusion

By demonstrating that fish can generate a neck-like motion through dorsoventral flexion across many intervertebral joints, this study opens a new perspective on the morphology, mechanics and evolution of fish vertebrae as both feeding and swimming structures with three-dimensional mobility. Relative to tetrapods, trout and frogfish present an alternative mechanism for moving the head relative to the body: using large regions of the vertebral column and not just the cranial-most region. Investigations are now needed to understand the mechanisms and evolution of this dorsoventral vertebral flexion in ray-finned and lobe-finned fishes.

Ethics. All animal procedures were done under Home Office Licence (P3658DB2) and followed protocols approved by the University of Liverpool Animal Welfare and Ethics Review Board.

Data accessibility. The raw X-ray video and CT data and their essential metadata have been deposited in the XMAPortal (http://xmaportal.org/webportal/, permanent identifier ULIVERPOOL1 for rainbow trout and ULIVERPOOL2 for frogfish). Trout data used in this study can be accessed with this link: https://xmaportal.org/webportal/larequest.php?request=CollectionViewAllFiles&StudyID=1&instit=ULIVERPOOL&collectionID=1. Frogfish data used in this study can be accessed with this link: https://xmaportal.org/webportallarequest.php?request=CollectionView&StudyID=2&instit=ULIVERPOOL&collectionID=2.

Authors' contributions. A.L.C.: conceptualization, formal analysis, funding acquisition, investigation, methodology, project administration, visualization, writing-original draft, writing-review and editing

Competing interests. I declare I have no competing interests

Funding. This work was supported by the UKRI Biotechnology and Biological Sciences Research Council (grant no. Fellowship BB/R011109/1) and the US National Science Foundation (grant no. 1655756).

Acknowledgements. I thank R. Sweeney, V. Preston, B. Cotterill, and the Biomedical Services Unit staff for fish husbandry support; H. Amplo and K. Magee for advice on fish maintenance; J. Gardiner, J. Charles, A. Hamilton, R. Kissane and P. Falkingham for assistance with X-ray filming; J. Sharkey and H. Poptami at the Centre for Preclinical Imaging, University of Liverpool and

P. Falkingham for CT imaging support; S. Gatesy and P. Falkingham advice on three-dimensional animation; N. Jeffery, P. Falkingham, E. Brainerd, for reading drafts; the Brainerd Lab at Brown University and the Evolutionary Morphology and Biomechanics Lab at University of Liverpool for discussions; two anonymous reviewers for thoughtful and constructive feedback.

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
