## [Peer Review File · Proceedings of the Royal Society B: Biological Sciences]

Review History

RSPB-2021-1091.R0 (Original submission)

Review form: Reviewer 1

Recommendation

Major revision is needed (please make suggestions in comments)

Scientific importance: Is the manuscript an original and important contribution to its field?

Excellent

General interest: Is the paper of sufficient general interest?

Excellent

Quality of the paper: Is the overall quality of the paper suitable?

Excellent

Is the length of the paper justified?

No

Should the paper be seen by a specialist statistical reviewer?

No

Do you have any concerns about statistical analyses in this paper? If so, please specify them explicitly in your report.

No

It is a condition of publication that authors make their supporting data, code and materials available - either as supplementary material or hosted in an external repository. Please rate, if applicable, the supporting data on the following criteria.

Is it accessible?

Yes

Is it clear?

Yes

Is it adequate?

Yes

Do you have any ethical concerns with this paper?

No

Comments to the Author

I commend the author on starting what I hope will be more study of the functional anatomy of the anterior vertebrae in fishes. This region of the vertebral column is highly variable across fishes and thus likely plays different roles depending on the group. This paper uses a sophisticated technique to analyze the movement of the anterior vertebrae during feeding in 2 quite distinct fish demonstrating that there are d-v intervertebral movements during feeding.

Lines 37-38: While the majority of actinopterygian fish have a connection between the pectoral girdle and the skull, this is not true for all actinopterygian fishes or for any cartilaginous fishes. It would be good to include some nuance here given this anatomical diversity.

Lines 93-94: Were only successful strikes included? Also, how often did strikes include cranial elevation less than 5 degrees? Is there a cranial elevation that did not coincide with vertebral movement? I am trying to get an understanding of how often the vertebral column plays in feeding in these two animals.

The author sets up how fish don't have an anatomical neck due to the connection between the pectoral girdle and skull. However, much of the vertebral movement appears to be posterior to this location. While the author does allude to the functional neck movements being a significant part of the anterior vertebral column, this isn't brought into context with the placement of the pectoral girdle. Would you expect more intervertebral movement in species with no connection of the pectoral girdle to the skull?

It would be great for the author to discuss how the movements of the vertebrae are possibly being controlled given that the myomeres responsible for moving these vertebrae span multiple segments. What does the muscular anatomy look like in this region of the body?

Review form: Reviewer 2

Recommendation

Major revision is needed (please make suggestions in comments)

Scientific importance: Is the manuscript an original and important contribution to its field?

Good

General interest: Is the paper of sufficient general interest?

Good

Quality of the paper: Is the overall quality of the paper suitable?

Good

Is the length of the paper justified?

Yes

Should the paper be seen by a specialist statistical reviewer?

No

Do you have any concerns about statistical analyses in this paper? If so, please specify them explicitly in your report.

No

It is a condition of publication that authors make their supporting data, code and materials available - either as supplementary material or hosted in an external repository. Please rate, if applicable, the supporting data on the following criteria.

Is it accessible?

Yes

Is it clear?

Yes

Is it adequate?

Yes

Do you have any ethical concerns with this paper?

No

Comments to the Author

This is the first in-depth study of in-vivo vertebral kinematics during suction feeding in fish. Consequently, the manuscript presents very interesting and novel kinematic data. It can also provide the basis for further research into the mechanics and form-function relationships of the vertebral column during suction feeding. The text is clearly written and nicely illustrated in the figures and supplementary videos. Methodologically, I have no concerns as the author is a leading expert in x-ray-based kinematic analyses.

My main criticisms is about the broader framing of the study: the fish vs tetrapods comparison is not clear to me. This concerns the description of the observed motions to be neck-like or the fish's vertebral column to have a functional neck as in tetrapods. As I understood, any vertebra performing some degree of rotation in the sagittal plane during the act of suction feeding are interpreted to be perform a neck-like function. What are the arguments for this view? I'm missing this information, as the function of a neck during feeding in tetrapods has not been defined properly in the text. In terrestrial lower vertebrates like lizards, the neck allows the head and jaws to be aimed and moved towards the food in 3D involving both pitch, yaw, and roll, while the trunk remains stationary (e.g. studies by S. Montuelle). If this would be the reference for neck-like function in a tetrapod, I see more differences than similarities with the current results. Namely, the vertebral column in these fish moves in response to a short blast of high-power muscle activity during the strike at the food, while the lizards manage to stably hold the neck into these 3D postures when approaching the food. So at least the temporal and dynamic aspects of these motions do not seem 'neck-like' with respect to lizards.

I fully agree that the vertebral column has an important role for head movement during suction feeding. Yet, the author concludes that the vertebral column translates and positions the head, but did not consider the vertebral column to move in response to the dynamics of the head.

Especially for the interpretation of the data of *Antennarius*, this is important. This animal is experiencing a strong external force in the form of suction propulsion. This forward pulling of the head may cause the straightening of the curves spine. Additionally, the clockwise torque on the head will inevitably push down on the first vertebra, an effect that has also been noted in seahorses (ref 29 in the manuscript; and note examples like the fused four cranialmost vertebra of the powerful head rotator *Fistularia* seems a response to avoid a collapse of this vertebral region). With this in mind, I'm unsure whether claiming that the data show that the vertebra are translating and positioning the head is appropriate. In other words, I'm afraid that readers will interpret this as an active process driven entirely from within the vertebral musculoskeletal system (statements such as line 59: 'vertebral column producing neck-like motions'), as during slow neck movement in humans, while this is not the case.

Minor remarks:

- (1) Line 26: word 'of' should be deleted.
- (2) Line 27: final sentence - has been written quite a few times now for the muscles surrounding these vertebra and the pectoral girdle. Are you sure you want to repeat this once more, claiming a 'recast' of our scientific view to this dual function?
- (3) Line 37: 'pectoral girdle attaches directly to the cranium'. There may be some exceptions, like eels. How was it for the shark in your previous publication in Proc. R. Soc B?
- (4) Line 126 and other places in the manuscript: 'dorsoventral rotations'. As 'dorsovertral' is a lineal axis, this may be less appropriate than 'dorsoventral plane rotations' or 'sagittal plane rotations'.
- (5) Line 179: 'used large regions of the vertebrae', I think you mean 'used large regions of the vertebral column'.

Decision letter (RSPB-2021-1091.R0)

28-Jun-2021

Dear Dr Camp:

Your manuscript has now been peer reviewed and the reviews have been assessed by an Associate Editor. Based on my own read of your manuscript as well as the assessments of the reviewers and the AE, I find your article to be of potential interest for Proceedings B. However, I agree with the reviewers that there are two major issues that must be addressed prior to further consideration. First, and most importantly, whereas the results are impressive, your study's impact relies on the comparison between the tetrapod neck and the vertebral movement in the fishes. Right now this is not entirely clear, and needs to be better articulated to ensure that the reader understands the impact of your findings. Second, as both reviewers also point out, there needs to be some discussion how neck movement is controlled. Both reviewers' comments and those of the AE are included at the end of this email for your reference.

Research ethics:

Use of animals and field studies:

It is a condition of publication that you make available the data and research materials supporting the results in the article. Please see our Data Sharing Policies (<https://royalsociety.org/journals/authors/author-guidelines/#data>). Datasets should be deposited in an appropriate publicly available repository and details of the associated accession number, link or DOI to the datasets must be included in the Data Accessibility section of the article (<https://royalsociety.org/journals/ethics-policies/data-sharing-mining/>). Reference(s) to datasets should also be included in the reference list of the article with DOIs (where available).

Please submit a copy of your revised paper within three weeks. If we do not hear from you within this time your manuscript will be rejected. If you are unable to meet this deadline please let us know as soon as possible, as we may be able to grant a short extension.

Best wishes,
Dr Sarah Brosnan
Editor, Proceedings B
mailto:proceedingsb@royalsociety.org

Associate Editor

Comments to Author:

Both reviewers give the manuscript high marks in general and agree that the data are very valuable, but they raise concerns about the framing and context provided. Most critically, Reviewer 2 questions the functional comparisons between fish and tetrapod necks. What is meant by "neck-like function" should be better defined. In addition, both reviewers request a better explanation of how the head movements are being produced and controlled: by muscular activity? By external forces? Expanding the discussion to address the musculoskeletal anatomy and biomechanical environment of these fish in detail would help to provide a basis for future research in this area.

Reviewer(s)' Comments to Author:

Referee: 1

Comments to the Author(s)

I commend the author on starting what I hope will be more study of the functional anatomy of the anterior vertebrae in fishes. This region of the vertebral column is highly variable across fishes and thus likely plays different roles depending on the group. This paper uses a sophisticated technique to analyze the movement of the anterior vertebrae during feeding in 2 quite distinct fish demonstrating that there are d-v intervertebral movements during feeding.

Lines 37-38: While the majority of actinopterygian fish have a connection between the pectoral girdle and the skull, this is not true for all actinopterygian fishes or for any cartilaginous fishes. It would be good to include some nuance here given this anatomical diversity.

Lines 93-94: Were only successful strikes included? Also, how often did strikes include cranial elevation less than 5 degrees? Is there a cranial elevation that did not coincide with vertebral movement? I am trying to get an understanding of how often the vertebral column plays in feeding in these two animals.

The author sets up how fish don't have an anatomical neck due to the connection between the pectoral girdle and skull. However, much of the vertebral movement appears to be posterior to this location. While the author does allude to the functional neck movements being a significant part of the anterior vertebral column, this isn't brought into context with the placement of the pectoral girdle. Would you expect more intervertebral movement in species with no connection of the pectoral girdle to the skull?

It would be great for the author to discuss how the movements of the vertebrae are possibly being controlled given that the myomeres responsible for moving these vertebrae span multiple segments. What does the muscular anatomy look like in this region of the body?

Referee: 2

Comments to the Author(s)

This is the first in-depth study of in-vivo vertebral kinematics during suction feeding in fish. Consequently, the manuscript presents very interesting and novel kinematic data. It can also provide the basis for further research into the mechanics and form-function relationships of the vertebral column during suction feeding. The text is clearly written and nicely illustrated in the figures and supplementary videos. Methodologically, I have no concerns as the author is a leading expert in x-ray-based kinematic analyses.

My main criticism is about the broader framing of the study: the fish vs tetrapods comparison is not clear to me. This concerns the description of the observed motions to be neck-like or the fish's vertebral column to have a functional neck as in tetrapods. As I understood, any vertebra performing some degree of rotation in the sagittal plane during the act of suction feeding are interpreted to be perform a neck-like function. What are the arguments for this view? I'm missing this information, as the function of a neck during feeding in tetrapods has not been defined properly in the text. In terrestrial lower vertebrates like lizards, the neck allows the head and jaws to be aimed and moved towards the food in 3D involving both pitch, yaw, and roll, while the trunk remains stationary (e.g. studies by S. Montuelle). If this would be the reference for neck-like function in a tetrapod, I see more differences than similarities with the current results. Namely, the vertebral column in these fish moves in response to a short blast of high-power muscle activity during the strike at the food, while the lizards manage to stably hold the neck into these 3D postures when approaching the food. So at least the temporal and dynamic aspects of these motions do not seem 'neck-like' with respect to lizards.

I fully agree that the vertebral column has an important role for head movement during suction feeding. Yet, the author concludes that the vertebral column translates and positions the head, but did not consider the vertebral column to move in response to the dynamics of the head. Especially for the interpretation of the data of *Antennarius*, this is important. This animal is experiencing a strong external force in the form of suction propulsion. This forward pulling of the head may cause the straightening of the curved spine. Additionally, the clockwise torque on the head will inevitably push down on the first vertebra, an effect that has also been noted in seahorses (ref 29 in the manuscript; and note examples like the fused four cranialmost vertebra of the powerful head rotator *Fistularia* seems a response to avoid a collapse of this vertebral region). With this in mind, I'm unsure whether claiming that the data show that the vertebra are translating and positioning the head is appropriate. In other words, I'm afraid that readers will interpret this as an active process driven entirely from within the vertebral musculoskeletal system (statements such as line 59: 'vertebral column producing neck-like motions'), as during slow neck movement in humans, while this is not the case.

Minor remarks:

- (1) Line 26: word 'of' should be deleted.
- (2) Line 27: final sentence – has been written quite a few times now for the muscles surrounding these vertebra and the pectoral girdle. Are you sure you want to repeat this once more, claiming a 'recast' of our scientific view to this dual function?
- (3) Line 37: 'pectoral girdle attaches directly to the cranium'. There may be some exceptions, like eels. How was it for the shark in your previous publication in Proc. R. Soc B?
- (4) Line 126 and other places in the manuscript: 'dorsoventral rotations'. As 'dorsovertral' is a lineal axis, this may be less appropriate than 'dorsoventral plane rotations' or 'sagittal plane rotations'.
- (5) Line 179: 'used large regions of the vertebrae', I think you mean 'used large regions of the vertebral column'.

Author's Response to Decision Letter for (RSPB-2021-1091.R0)

See Appendix A.

RSPB-2021-1091.R1 (Revision)

Review form: Reviewer 1

Recommendation

Accept as is

Scientific importance: Is the manuscript an original and important contribution to its field?

Good

General interest: Is the paper of sufficient general interest?

Good

Quality of the paper: Is the overall quality of the paper suitable?

Excellent

Is the length of the paper justified?

Yes

Should the paper be seen by a specialist statistical reviewer?

No

Do you have any concerns about statistical analyses in this paper? If so, please specify them explicitly in your report.

No

It is a condition of publication that authors make their supporting data, code and materials available - either as supplementary material or hosted in an external repository. Please rate, if applicable, the supporting data on the following criteria.

Is it accessible?

Yes

Is it clear?

Yes

Is it adequate?

Yes

Do you have any ethical concerns with this paper?

No

Comments to the Author

I appreciate the author's willingness to consider a reframing of the issues brought by the reviewers and editor. I have no other major comments to make or concerns about the way the comments were addressed. The only additional comment I have is that I think there is a missing word in Line 359. I think it should read "joints in extant and fossil fishes".

Review form: Reviewer 2

Recommendation

Accept with minor revision (please list in comments)

Scientific importance: Is the manuscript an original and important contribution to its field?

Good

General interest: Is the paper of sufficient general interest?

Good

Quality of the paper: Is the overall quality of the paper suitable?

Excellent

Is the length of the paper justified?

Yes

Should the paper be seen by a specialist statistical reviewer?

No

Do you have any concerns about statistical analyses in this paper? If so, please specify them explicitly in your report.

No

It is a condition of publication that authors make their supporting data, code and materials available - either as supplementary material or hosted in an external repository. Please rate, if applicable, the supporting data on the following criteria.

Is it accessible?

Yes

Is it clear?

Yes

Is it adequate?

Yes

Do you have any ethical concerns with this paper?

No

Comments to the Author

My comments have been addressed in this revision. The authors asked for more information on vertebral fusion in *Fistularidae*, which is most likely an adaptation to withstand the high reaction forces from powerful cranial elevation. Here are two links:

https://monarch.calacademy.org/mnt/image_db/ichtypes/radio/hi/106808-r.jpg

<https://doi.org/10.1007/s00435-019-00470-4>

This may be worth to mention, e.g. that the vertebral column also serves a function to withstand the strong reaction forces that result from pulling on the head by the epaxials.

Decision letter (RSPB-2021-1091.R1)

02-Aug-2021

Dear Dr Camp

I am pleased to inform you that your manuscript RSPB-2021-1091.R1 entitled "A neck-like vertebral motion in fish" has been accepted for publication in Proceedings B.

The referee(s) have recommended publication, but also suggest some minor revisions to your manuscript. Therefore, I invite you to respond to the referee(s)' comments as you see fit. Because the schedule for publication is very tight, it is a condition of publication that you submit the revised version of your manuscript within 7 days. If you do not think you will be able to meet this date please let us know.

Sincerely,
Dr Sarah Brosnan
Editor, Proceedings B
<mailto:proceedingsb@royalsociety.org>

Associate Editor:
Board Member: 1
Comments to Author:

The reviewers are satisfied that all concerns have been addressed. Reviewer 1 provides two additional references that the author may want to include.

Reviewer(s)' Comments to Author:

Referee: 2

Comments to the Author(s)

My comments have been addressed in this revision. The authors asked for more information on vertebral fusion in Fistularidae, which is most likely an adaptation to withstand the high reaction forces from powerful cranial elevation. Here are two links:

https://monarch.calacademy.org/mnt/image_db/ichtypes/radio/hi/106808-r.jpg

<https://doi.org/10.1007/s00435-019-00470-4>

This may be worth to mention, e.g. that the vertebral column also serves a function to withstand the strong reaction forces that result from pulling on the head by the epaxials.

Referee: 1

Comments to the Author(s)

I appreciate the author's willingness to consider a reframing of the issues brought by the reviewers and editor. I have no other major comments to make or concerns about the way the comments were addressed. The only additional comment I have is that I think there is a missing word in Line 359. I think it should read "joints in extant and fossil fishes".

Author's Response to Decision Letter for (RSPB-2021-1091.R1)

See Appendix B.

Decision letter (RSPB-2021-1091.R2)

04-Aug-2021

Dear Dr Camp

I am pleased to inform you that your manuscript entitled "A neck-like vertebral motion in fish" has been accepted for publication in Proceedings B.

Your article has been estimated as being 7 pages long. Our Production Office will be able to confirm the exact length at proof stage.

Data Accessibility section

Open Access

Paper charges

Sincerely,
Editor, Proceedings B
<mailto:proceedingsb@royalsociety.org>

Appendix A

Response to Reviewers

Thank you for these helpful and thoughtful responses, and especially the discussion of different mechanisms for producing cranial elevation. I have addressed all the reviewers' and editors' comments and concerns (see below) and in almost all cases implemented their suggested changes in my manuscript (see track-changes document). Below I have copied the original comments (in italics) with my responses and descriptions of the corresponding revisions (in bold). This is followed by a copy of the manuscript with revisions marked as 'tracked changes'.

Associate Editor

1. Both reviewers give the manuscript high marks in general and agree that the data are very valuable, but they raise concerns about the framing and context provided. Most critically, Reviewer 2 questions the functional comparisons between fish and tetrapod necks. What is meant by "neck-like function" should be better defined. In addition, both reviewers request a better explanation of how the head movements are being produced and controlled: by muscular activity? By external forces? Expanding the discussion to address the musculoskeletal anatomy and biomechanical environment of these fish in detail would help to provide a basis for future research in this area.

1. I am delighted the editor and reviewers found these data valuable. I have revised the framing and context of the study in response to the reviewers' concerns:

I now explicitly define the neck-like function this study is considering as dorsoventral flexion of the craniovertebral/intervertebral joints to rotate the head (lines 44-45). And I focus my comparison between fish and tetrapods specifically on the proportion of the vertebral column used to achieve this function (lines 55-61). While tetrapods rely on a cranial region of specialized vertebrae (the neck), trout and frogfish had dorsoventral rotations across large regions of the vertebral column—far beyond the morphologically distinct cervical vertebrae (185-187). These results provide a starting point for investigating how other non-tetrapod vertebrates may use this neck-like, dorsoventral rotation of the vertebral column to move the head relative to the body (lines 233-243).

I agree that how these vertebral motions are produced and controlled is an important topic and added a discussion of possible mechanisms (lines 204-220). These include active shortening of the dorsal body (epaxial) muscles to rotate the head, rotate the vertebrae, or (more likely) both, and the external fluid forces generated by suction feeding (see response #10-11 for more details). And I have provided a brief summary of the notoriously complex epaxial muscle anatomy (lines 207-213) and how it may relate to these vertebral motions. Conclusively determining which mechanisms are used by trout and frogfish and how the complex architecture of the segmented epaxial muscles produce these motions is beyond the scope of this study. This would require substantial new data such epaxial length changes during cranial elevation, and detailed epaxial anatomy--which to my knowledge has never been described in frogfish. However, I have (hopefully!) provided enough information to highlight specific areas of this fascinating system where future research is needed (lines 212-213) and provide a foundation for that work.

Referee: 1

Comments to the Author(s)

2. I commend the author on starting what I hope will be more study of the functional anatomy

of the anterior vertebrae in fishes. This region of the vertebral column is highly variable across fishes and thus likely plays different roles depending on the group. This paper uses a sophisticated technique to analyze the movement of the anterior vertebrae during feeding in 2 quite distinct fish demonstrating that there are d-v intervertebral movements during feeding.

2. Thank you, I also hope to continue studying the functional morphology of these anterior vertebrae in fishes!

3. Lines 37-38: While the majority of actinopterygian fish have a connection between the pectoral girdle and the skull, this is not true for all actinopterygian fishes or for any cartilaginous fishes. It would be good to include some nuance here given this anatomical diversity.

3. I agree completely, and it was certainly not my intention to overlook the remarkable diversity of pectoral girdle anatomy across bony and cartilaginous fishes. I have changed the text here to specify that I am referring to ray-finned fishes and that the pectoral girdle articulates with the skull in most of these fish (lines 37-40). I also now mention the very different pectoral girdle anatomy of sharks in discussing how my results may inform future studies of other groups of fish (lines 236-238).

4. Lines 93-94: Were only successful strikes included? Also, how often did strikes include cranial elevation less than 5 degrees? Is there a cranial elevation that did not coincide with vertebral movement? I am trying to get an understanding of how often the vertebral column plays in feeding in these two animals.

4. I have provided more detail on the strikes chosen for analysis in the methods (lines 95-97). All the frogfish and 18 of the 20 trout strikes analysed were successful, which probably reflects the relatively unchallenging prey items: mealworms, pellets, and dead fish or shrimp. I did include two unsuccessful misses from one trout: these were kinematically similar to the successful strikes with relatively large (for these trout) cranial elevations of 10-12 degrees. All frogfish strikes and most (20 out of 30) trout strikes had at least 5 degrees of cranial elevation. I never observed cranial elevation resulting from just rotation of the craniovertebral joint: it was always accompanied by dorsoventral motion of multiple intervertebral joints.

My personal observation is that trout relied less on dorsoventral vertebral rotation and cranial elevation to capture food—this is reflected in the highly variable magnitudes and patterns of these motions (Figs. 2-3 and Supplemental Fig. 3). These fish are excellent swimmers and use both rapid mouth expansion (including cranial elevation) to suck food into the mouth, as well as forward swimming to propel the mouth towards the food (ram feeding). And they adjusted the position and orientation of their whole body in the tank, relative to the food.

In contrast, I suspect vertebral motion is essential for feeding in frogfish. Cranial and vertebral kinematics were surprisingly consistent, compared to trout (Figs. 2-4). These sit-and-wait, benthic predators locomote slowly and rely on massive and rapid mouth expansion to exert suction forces and protrude the jaws towards the food. Dorsoventral vertebral rotation to elevate the head not only directly expands the mouth cavity dorsally, but can contribute to ventral and lateral expansion and jaw protrusion through musculoskeletal linkages in the fish head. And in frogfish the direction of vertebral motion seems linked to the position of the head and mouth relative to the food (Supplemental Fig. 2), rather than moving and re-orienting the whole body like trout.

Of course, these observations come with the caveat that they are based on 3 individuals, in an artificial lab setting, feeding on non-elusive food items.

5. *The author sets up how fish don't have an anatomical neck due to the connection between the pectoral girdle and skull. However, much of the vertebral movement appears to be posterior to this location. While the author does allude to the functional neck movements being a significant part of the anterior vertebral column, this isn't brought into context with the placement of the pectoral girdle. Would you expect more intervertebral movement in species with no connection of the pectoral girdle to the skull?*

5. I agree with the reviewer that the role of pectoral girdle position in neck-like vertebral motions was unclear. It was not my intention to imply that a connection between the skull and pectoral girdle limits intervertebral motion in fish. I now describe the differences in pectoral girdle placement between ray-finned fish and tetrapods in more detail (lines 38-40). Motion between the head and pectoral girdle in ray-finned fish is determined by the joints linking these structures. This is quite different from tetrapods, where the vertebrae of the neck link the head and pectoral girdle. And it suggests that in the “neck” of ray-finned fish, cranio-pectoral mobility may be independent from cranio-vertebral mobility. Because of this, I expect intervertebral movement could be relatively independent of how tightly or loosely the pectoral girdle is connected to the skull, so species without this connection may not necessarily have more vertebral mobility.

Also, I now state that the dorsoventral rotations of the vertebral column extend far beyond the pectoral girdle (line 181), and have added the pectoral girdle to Fig. 1 to show its position relative to the vertebral column. While investigating the motion of the pectoral girdle relative to the skull and vertebral column is beyond the scope of this study, it is an intriguing question that I hope will be tackled in future work.

6. *It would be great for the author to discuss how the movements of the vertebrae are possibly being controlled given that the myomeres responsible for moving these vertebrae span multiple segments. What does the muscular anatomy look like in this region of the body?*

6. I agree that the muscular system is an important and exciting component of vertebral motions, and have added a discussion of the possible mechanisms by which epaxial muscles could generate dorsoventral rotation of the head and vertebral column (lines 204-213). Possibly, fish are using different amounts of muscle activation and shortening within a single myomere as in Ellerby and Altringham, 2001. Much is still unknown and exactly how fish use the axial muscles to generate feeding and swimming motions is still an active research area (e.g., Jimenez and Brainerd, 2020; Jimenez et al., 2021).

A full description of the complex anatomy of the epaxial myomeres is beyond the scope of this study (and is currently unknown in frogfish, as far as I know). Previous studies have focused primarily on the swimming function of these muscles and mostly examined the midbody or caudal regions. Craniocaudal gradients in myomere shape—changes in the relative proportions of the different “cones” of the myomere--have been noted in some species including other salmonids. Comparing the morphology of the epaxial muscles relative to the location and magnitude of dorsoventral vertebral rotation would be an exciting and interesting future study.

Jimenez, Y. E., Brainerd, E. L., 2020. Dual function of epaxial musculature for swimming and suction feeding in largemouth bass. Proc Biol Sci 287, 20192631, doi:10.1098/rspb.2019.2631.

Jimenez, Y. E., Marsh, R. L., Brainerd, E. L., 2021. A biomechanical paradox in fish: swimming and suction feeding produce orthogonal strain gradients in the axial musculature. *Sci Rep* 11, 10334, doi:10.1038/s41598-021-88828-x.

Referee: 2

Comments to the Author(s)

7. This is the first in-depth study of in-vivo vertebral kinematics during suction feeding in fish. Consequently, the manuscript presents very interesting and novel kinematic data. It can also provide the basis for further research into the mechanics and form-function relationships of the vertebral column during suction feeding. The text is clearly written and nicely illustrated in the figures and supplementary videos. Methodologically, I have no concerns as the author is a leading expert in x-ray-based kinematic analyses.

7. Thank you! I'm glad you found the data interesting and the writing and images clear and helpful.

8. My main criticisms is about the broader framing of the study: the fish vs tetrapods comparison is not clear to me. This concerns the description of the observed motions to be neck-like or the fish's vertebral column to have a functional neck as in tetrapods. As I understood, any vertebra performing some degree of rotation in the sagittal plane during the act of suction feeding are interpreted to be perform a neck-like function. What are the arguments for this view? I'm missing this information, as the function of a neck during feeding in tetrapods has not been defined properly in the text. In terrestrial lower vertebrates like lizards, the neck allows the head and jaws to be aimed and moved towards the food in 3D involving both pitch, yaw, and roll, while the trunk remains stationary (e.g. studies by S. Montuelle). If this would be the reference for neck-like function in a tetrapod, I see more differences than similarities with the current results. Namely, the vertebral column in these fish moves in response to a short blast of high-power muscle activity during the strike at the food, while the lizards manage to stably hold the neck into these 3D postures when approaching the food. So at least the temporal and dynamic aspects of these motions do not seem 'neck-like' with respect to lizards.

8. I agree that the original manuscript did not make it clear exactly what functions and motions of the neck were being compared between fish and tetrapods. I now specify that this study is focusing on just one function of the neck, namely the ability to rotate the head dorsoventrally in the sagittal plane, relative to the body (lines 47-48). This is consistent with previous definitions of a 'functional neck' and 'neck-bending' in fish (Ford, 1937; Lesiuk and Lindsey, 1978; Lauder and Liem, 1981; Johanson et al., 2003) and is considered key function of the tetrapod neck often used in feeding (Heiss et al., 2018; Shubin et al., 2015). Throughout the manuscript, I have removed the phrase 'neck-like motion' or clarified it with more specific descriptions of dorsoventral rotation of the head and/or vertebral column. And I have modified the title to reflect that this study focuses on only a single motion (line 1).

Additionally, I focus the comparison between fish and tetrapods on what portion of the vertebral column is used to produce dorsoventral head rotation (lines 63-71). This includes highlighting how trout and frogfish differ from tetrapods: rather than using solely the cranial-most, "cervical" vertebrae—analogous to the tetrapod neck—they rotate a much larger region of the vertebral column during cranial elevation (line 185-187). This distinction supports the idea that fish use large regions of the vertebral column for both feeding and swimming, as opposed to using just cranial vertebrae for dorsoventral rotation in feeding and just caudal vertebrae for lateral rotation in

swimming. And it suggests that future studies investigating “functional necks” in non-tetrapods should consider more caudal vertebrae and not just the craniovertebral joint (lines 318-319).

Lesiuk, T., Lindsey, C., 1978. Morphological peculiarities in neck-bending amazonian Characoid fish *Rhaphiodon vulpinus*. *Can J Zool* 56, 991-997.

Lauder, G., Liem, K. F., 1981. Prey capture by *Luciocephalus pulcher*: implications for models of jaw protrusion in teleost fishes. *Environmental Biology of Fishes* 6, 257-268.

9. I fully agree that the vertebral column has an important role for head movement during suction feeding. Yet, the author concludes that the vertebral column translates and positions the head, but did not consider the vertebral column to move in response to the dynamics of the head. Especially for the interpretation of the data of *Antennarius*, this is important. This animal is experiencing a strong external force in the form of suction propulsion. This forward pulling of the head may cause the straightening of the curves spine.

9. This is an excellent point and particularly helpful for thinking about how frogfish achieve such a dramatic change in vertebral posture. I have added a discussion of the possible role of external fluid forces in producing craniovertebral motion (lines 215-220). Interestingly, most of the power for mouth expansion--and therefore these external fluid forces—is likely generated by the epaxial muscles, suggesting they are both directly and indirectly contributing to dorsoventral vertebral flexion.

10. Additionally, the clockwise torque on the head will inevitably push down on the first vertebra, an effect that has also been noted in seahorses (ref 29 in the manuscript; and note examples like the fused four cranialmost vertebra of the powerful head rotator *Fistularia* seems a response to avoid a collapse of this vertebral region). With this in mind, I'm unsure whether claiming that the data show that the vertebra are translating and positioning the head is appropriate. In other words, I'm afraid that readers will interpret this as an active process driven entirely from within the vertebral musculoskeletal system (statements such as line 59: 'vertebral column producing neck-like motions'), as during slow neck movement in humans, while this is not the case.

10. I agree that these data show how the neurocranium and vertebral column are moving, but cannot determine the mechanisms driving these motions. I have revised the text throughout the manuscript to avoid implying that cranial motion is entirely produced by activation of the vertebral musculoskeletal system. And in discussing how these vertebral motions could be generated, I explicitly state that the epaxials could be acting on the head to rotate the vertebrate or vice versa or (what I suspect is most likely) some combination of both (lines 205-207). I have cited the seahorse system as an example (Van Wassenbergh et al., 2011), but was unable to find references for the cranial rotation and morphology of *Fistularia*. If the reviewer can suggest any literature on this species that I may have missed, I would be grateful and happy to reference them.

Minor remarks:

11. Line 26: word 'of' should be deleted.

11. Thank you for spotting this error, I have fixed it.

12. Line 27: final sentence – has been written quite a few times now for the muscles surrounding these vertebra and the pectoral girdle. Are you sure you want to repeat this once more, claiming a 'recast' of our scientific view to this dual function?

12. I appreciate the reviewer's invitation to improve this concluding sentence. I have revised lines 27-29 to emphasize the importance of large regions of the vertebral column--not just the cranial-most vertebrae—for producing neck-like cranial elevation in fish and potentially other non-tetrapods.

13. Line 37: *'pectoral girdle attaches directly to the cranium'*. *There may be some exceptions, like eels. How was it for the shark in your previous publication in Proc. R. Soc B?*

13. I agree completely and now specify that this statement refers to most, ray-finned fishes (lines 37-39). And I discuss the different position of the pectoral girdle in sharks (line 237-238).

14. Line 126 and other places in the manuscript: *'dorsoventral rotations'*. *As 'dorsovertral' is a lineal axis, this may be less appropriate than 'dorsoventral plane rotations' or 'sagittal plane rotations'*.

14. I have chosen to keep the term 'dorsoventral rotation' to be consistent with how this craniovertebral motion is described in the literature for tetrapods and fish (e.g., Heiss et al., 2018; Camp, 2019; Kambic et al., 2017). But I have clarified at several points in the manuscript that this refers to rotation in the sagittal plane (lines 51, 56, 167).

Kambic, R. E., Biewener, A. A., Pierce, S. E., 2017. Experimental determination of three-dimensional cervical joint mobility in the avian neck. *Frontiers in Zoology* 14, doi:10.1186/s12983-017-0223-z.

15. Line 179: *'used large regions of the vertebrae'*, I think you mean *'used large regions of the vertebral column'*.

15. Thank you, I have corrected this.

Appendix B

Response to Referees

I am delighted that both referees are satisfied with the revised manuscripts and do not have any additional concerns. Their constructive and thoughtful feedback has certainly improved this study. I have addressed both of their remaining comments, implementing their suggested changes in my manuscript. Below I have copied the original comments (in italics) with my responses and descriptions of the corresponding revisions (in bold). This is followed by a copy of the manuscript with revisions marked as ‘tracked changes’.

Associate Editor

The reviewers are satisfied that all concerns have been addressed. Reviewer 1 provides two additional references that the author may want to include.

I have incorporated the additional ideas and references provided by Referee 2.

Referee 1

I appreciate the author's willingness to consider a reframing of the issues brought by the reviewers and editor. I have no other major comments to make or concerns about the way the comments were addressed. The only additional comment I have is that I think there is a missing word in Line 359. I think it should read “joints in extant and fossil fishes”.

I appreciate your careful proofreading, and I have corrected this sentence as suggested (line 250).

Referee 2

*My comments have been addressed in this revision. The authors asked for more information on vertebral fusion in *Fistularidae*, which is most likely an adaptation to withstand the high reaction forces from powerful cranial elevation. Here are two links:*

https://monarch.calacademy.org/mnt/image_db/ichtypes/radio/hi/106808-r.jpg

<https://doi.org/10.1007/s00435-019-00470-4>

This may be worth to mention, e.g. that the vertebral column also serves a function to withstand the strong reaction forces that result from pulling on the head by the epaxials.

Thank you for highlighting this paper on trumpetfish, which I had unintentionally overlooked. It is a fantastic example of a very different mechanism of using vertebral and body motions for feeding, and would be fascinating to explore in future studies. I now cite this study and others as examples of fish with both large cranial elevation and fused anterior vertebrae (Discussion: lines 228-234), and speculate that these fused vertebrae may function to resist the forces imposed by powerful cranial elevation. This raises quite an intriguing idea that both extreme increases and decreases in intervertebral joint mobility can contribute to feeding functions in fish.